# Athermal Concentration of Blueberry Juice by Forward Osmosis: Food Additives as Draw Solution

**DOI:** 10.3390/membranes12080808

**Published:** 2022-08-21

**Authors:** Haoqi Chu, Zhihan Zhang, Huazhao Zhong, Kai Yang, Peilong Sun, Xiaojun Liao, Ming Cai

**Affiliations:** 1Department of Food Science and Technology, Zhejiang University of Technology, Hangzhou 310014, China; 2Key Laboratory of Food Macromolecular Resources Processing Technology Research, Zhejiang University of Technology, China National Light Industry, Hangzhou 310014, China; 3Beijing Advanced Innovation Center for Food Nutrition and Human Health, College of Food Science and Nutritional Engineering, China Agricultural University, Beijing 100083, China; 4Beijing Key Laboratory for Food Nonthermal Processing, National Engineering Research Center for Fruit & Vegetable Processing, Beijing 100083, China

**Keywords:** forward osmosis, blueberry juice, food additive, draw solution, storage

## Abstract

This study is to evaluate the athermal forward osmosis (FO) concentration process of blueberry juice using food additives as a draw solution (DS). The effects of food additives, including citric acid, sodium benzoate, and potassium sorbate, on the concentration processes are studied, and their effects on the products and membranes are compared. Results show that all these three food additives can be alternative DSs in concentration, among which citric acid shows the best performance. The total anthocyanin content (TAC) of blueberry juice concentrated by citric acid, sodium benzoate, and potassium sorbate were 752.56 ± 29.04, 716.10 ± 30.80, and 735.31 ± 24.92 mg·L^−^^1^, respectively, increased by 25.5%, 17.8%, and 19.9%. Meanwhile, the total phenolic content (TPC) increased by 21.0%, 10.6%, and 16.6%, respectively. Citric acid, sodium benzoate, and potassium sorbate all might reverse into the concentrated juice in amounts of 3.083 ± 0.477, 1.497 ± 0.008, and 0.869 ± 0.003 g/kg, respectively. These reversed food additives can make the TPC and TAC in juice steadier during its concentration and storage. Accordingly, food additives can be an excellent choice for DSs in the FO concentration process of juices, not only improving the concentration efficiency but also increasing the stability of blueberry juice.

## 1. Introduction

The global market of concentrated juice, which is widely used to provide the ingredients for making the beverage, is estimated to be USD 81.8 billion in 2020. Concentration is a crucial step in concentrated juice production because concentrated juice may be stored for a long time and is portable [1]. Usually, juices were concentrated by thermal technologies, but nutrients in juice would be reduced by these technologies, particularly bioactive substances such as polyphenols, ascorbic acid, and anthocyanins [2,3,4]. It has been found that the retention rates of polyphenols and anthocyanins in pomegranate juice by thermal evaporation were lower than that by freezing crystallization [4] and the total phenolic content (TPC) of black mulberry juice decreased dramatically after thermal evaporation concentration [5]. Accordingly, athermal technology must be developed and applied to concentrate juices with thermally sensitive ingredients.

As a kind of athermal processing, membrane technologies, such as ultrafiltration (UF), nanofiltration (NF), and reverse osmosis (RO), have been applied to juice manufacture [6]. However, applications of RO and NF have been limited because of their high operating pressure and severe membrane fouling [7]. Recently, forward osmosis (FO) has become an alternative technique in juice concentration because of its exceptional quality of concentrated products under ambient pressure and temperature, as well as the energy savings it offers [8,9]. FO is a membrane-separation process driven by the osmotic pressure difference between the two sides of the selective permeation membrane to achieve spontaneous water transfer [10]. Compared with pressure-driven membrane separation processes such as UF, NF, and RO, FO inherently has many unique advantages, such as low-pressure- or even no-pressure operation and thus lower energy consumption [11]. Meanwhile, FO membranes are easy to clean due to their low fouling tendency [12]. FO has already been successfully applied in some juice-production cases. It was discovered that FO raised the total soluble solid (TSS) of pineapple juice and apple juice from 4.4 and 11 to 54 and 60 °Brix, respectively [13].

In FO processes, the draw solution (DS), the most important factor of such technology, was commonly used with inorganic salts such as sodium chloride, potassium chloride, magnesium chloride, and others [14]. These salts could easily reverse into the feed solution (FS), which led to negative impacts on the raw liquids, particularly on food matrices [15]. NaCl (11.0 g·m^−2^·h^−1^) exhibited a larger reverse salt penetration than glucose (2.5 g·m^−2^·h^−1^) in liquid egg white FO concentration, which could lessen the taste [16]. The shortcoming of the current study is the lack of a suitable DS that is stable, non-toxic, cheap, and protective. Fortunately, food additives have these characteristics. It was believed that picking a food additive as a DS might be good for the concentration of juices by FO because reverse salt permeation could not be avoided. 

Food additives, such as sodium benzoate, potassium sorbate, citric acid, sodium bisulfate, cyclamate, and sodium saccharin, are commonly used in fruit juices to reduce nutrient loss and prolong their shelf life [17,18,19]. As an acidic preservative, sodium benzoate performed best at a lower pH because of the higher content of undissociated benzoic acid, which is an active form of antimicrobial [20]. An acidized sodium benzoate solution was reported to be effective in reducing the population of *Escherichia coli* O157:H7, *Salmonella enterica*, and *Listeria monocytogenes* mixtures (>4 log CFU/g) on cherry tomatoes [21]. Similar to sodium benzoate, potassium sorbate and citric acid also have antibacterial effects in fruit juices [22]. Meanwhile, it has been demonstrated that potassium sorbate could be utilized as a DS in the FO process because of its high osmotic pressure [23]. However, this literature focused on the effects of food preservatives on the FO process. To the best of our knowledge, the effects of food additives on juices during the FO process and their storage have never been studied.

In this study, the effects of citric acid, potassium sorbate, and sodium benzoate on FO concentrations of blueberry juice were investigated. Differences in the water flux, membrane fouling, and reverse salt fluxes during FO were assessed. The effects of these food additives on the physicochemical properties and preservation of blueberry juice were also investigated. Meanwhile, the stabilities of blueberry juice concentrated by the FO process using different food additives were compared using a storage experiment.

## 2. Materials and Methods

### 2.1. Materials and Chemicals

Blueberry pulp was made from blueberries by a Juicer (SKG2098, Guangdong SKG Intelligent Technology Co., Ltd., Foshan, China). A total of 0.22% pectinase and 0.73% cellulase were added to the pulp and placed in a water bath (DK-S24, Shanghai SUMSUNG Laboratory Instrument Co., Ltd., Shanghai, China) at 50 °C for 100 min. After filtration with gauze, the clarified blueberry juice was centrifuged (L550-, Hunan Xiangyi Laboratory Instrument Development Co., Ltd., Changsha, China) at 4000 rpm for 15 min and stored in a refrigerator at 4 °C for further use as FS. Citric acid, potassium sorbate, and sodium benzoate of analytical reagent were purchased from Yien Chemical Technology Co., Ltd. (Shanghai, China).

A thin Film Composite (TFC) membrane with an effective area of 20 cm^2^ produced from polyamide served as an FO membrane, and was previously purchased from Guochu Science and Technology (Xiamen, China).

### 2.2. FO Concentration Process

The FO processes were carried out using a bench-scale FO system, as shown in Figure 1. A total of 150 mL of FS and 500 mL of DS were circulated by two peristaltic pumps at a counter-current cross-flow mode. The speed of peristaltic pumps on both sides was set at 150 rpm (flow rate of 120 mL·min^−1^), which was stable in preliminary experiments. FS tank was placed on a digital balance (STX622ZH, Ohaus Instruments (Changzhou) Co., Ltd., Changzhou, China) connected to a computer for water flux acquisition. The concentration process was carried out for 8 h, and the water flux was calculated according to Equation (1) [24].
J_W_ = Δm/(ρ × Δt × A)(1)
where J_W_ is the water flux (L·m^−2^·h^−1^), Δm is the mass change of feed liquid (g), ρ is the water density (1.0 g·cm^−3^), Δt is the concentration time (h), and A is the effective membrane area (m^2^).

### 2.3. Membrane-Fouling Analysis

The original and fouled membranes were dried naturally after FO processes. Dried membranes were cut into 0.5 × 0.5 cm^2^ pieces, sprayed gold, and observed through a field emission scanning electron microscope (FE-SEM, HITACHI Regulus 8100, Hitachi, Ltd., Tokyo, Japan) at an acceleration voltage of 15 kV.

### 2.4. Determination of Total Anthocyanin Content (TAC)

TAC was determined according to the pH differential method with slight modification [25]. To obtain the supernatant, 10 mL of blueberry juice was centrifuged at 4000 rpm for 15 min at a temperature of roughly 25 °C. A total of 1.0 mL of blueberry juice was siphoned into a 10 mL volumetric flask with a KCl buffer of pH 1.0 and sodium acetate buffer of pH 4.5, respectively. After standing for 2 h in the refrigerator at 4 °C, the absorbance value was determined by a UV spectrophotometer (GENESYS 150, Thermo Fisher Scientific Inc., Pittsburgh, PA, USA) at 520 nm and 700 nm. TAC was calculated according to Equation (2) (calculated by cyanidin-3-O-glycoside).
(2)$$\mathrm{TAC}(/\mathrm{mg}\cdot L{-}^{1})=\frac{\left[\left(A_{520}-{\;A}_{700}\right)\mathrm{pH}1.0-\left(A_{520}-{\;A}_{700}\right)\mathrm{pH}4.5\right]\times \;\mathrm{MW}\;\times \;\mathrm{DF}\;\times 1000}{\varepsilon \;\times \;L}$$
where MW is the molecular weight of cyanidin-3-O-glycoside (449.2 g·mol^−1^), ε is the molar extinction coefficient of cyanidin-3-O-glycoside (26,900 L·(mol·cm) ^−1^), and L is the optical path length (1 cm).

### 2.5. Determination of Clarity

According to the method described by Mondal et al. [26], the absorbance at 660 nm of juices was measured with distilled water as blank. The clarity of the sample was represented by T and calculated by Equation (3).
T = 100 × 10^−A^(3)

### 2.6. Determination of TPC

TPC was analyzed using a modified Folin–Ciocalteu colorimetric method [27]. A total of 1.0 mL blueberry juice was added to 5.0 mL Folin-phenol reagent with a 10% mass fraction into a test tube. After 5 min of reaction, 4.0 mL of a sodium carbonate solution with a mass fraction of 7.5% was added and mixed on a vortex oscillator. The absorbance value at 765 nm was measured using a gallic acid standard after standing at room temperature for 60 min. The standard equation was y = 0.0113x + 0.0174, R^2^ = 0.9996, 10–60 mg·L^−1^.

### 2.7. Determination of Total Sugar Content

Total sugar content was determined by a phenol sulfuric acid method [28]. A total of 1.0 mL of 6% phenol and 5.0 mL of concentrated sulfuric acid were added to 2.0 mL of blueberry juice. After mixing with a whirlpool oscillator, the juice was bathed in boiling water for 15 min. The absorbance of juice was measured at 490 nm as soon as it reached room temperature. The standard curve for glucose was established as an equation of y = 0.0114x + 0.0309, R^2^ = 0.9991, 10–60 mg·L^−1^.

### 2.8. Determination of pH, TSS, and Particle Size

The pH of blueberry juice was determined by a bench pH meter (ST3100, Ohaus Instruments (Changzhou) Co., Ltd., Changzhou, China). TSS was determined by an Abbe refractometer (ATC, Shanghai Zhuoguang Instrument Technology Co., Ltd., Shanghai, China). The particle size distribution of blueberry juice was determined by a laser particle size analyzer (Nano ZS, Malvern Instruments Co., Ltd., Worcestershire, UK).

### 2.9. Determination of Chromatic Aberration

The color difference analyzer (Color Quest XE, Hunter Associates Laboratory, Inc., Fairfax, VA, USA) was used according to a reported method [29].
(4)$$\Delta E=\sqrt{{\left(L^{*}-L^{0}\right)}^{2}+{\left(a^{*}-a^{0}\right)}^{2}+{\left(b^{*}-b^{0}\right)}^{2}}$$
where L* represents brightness; a* represents red intensity; b* represents yellow intensity; ΔE represents the total color change; and L^0^, a^0,^ and b^0^ represent the color values of blueberry juice.

### 2.10. Analysis of Anthocyanins by High-Performance Liquid Chromatography (HPLC)

HPLC system (LC-2030C 3D Plus, Shimadzu Enterprise Management Co., Ltd., Shanghai, China) equipped with UV detection was used to identify anthocyanins according to a developed method [30]. Before analysis, 1 mL of each sample was filtered with a 0.45 μm syringe filter. The separation was achieved using an Agilent Zorbax Eclipse Plus-C18 reverse column (5 μm, 4.6 mm × 250 mm) using 1% phosphoric acid as mobile phase A and acetonitrile as mobile phase B. Gradient program was set as 5%B (0–15 min), 10%B (15–35 min), 13%B (35–50 min), and 5%B (50–55 min). The temperature was controlled at 30 °C, and the UV detector was set at 520 nm. The injection volume was 10 μL, and the flow rate was 1.0 mL·min^−1^.

### 2.11. Determination of Food Additives in Concentrated Juice

The citric acid in blueberry juice was separated by HPLC on a C18 column (5 μm, 4.6 × 250 mm) using methanol−0.5% sodium dihydrogen phosphate (60:40) as the mobile phase [31]. The injection volume was 20 μL, and the flow rate was 1.0 mL·min^−1^. The column temperature was 25 °C, and the detection wavelength was 210 nm. Potassium sorbate in blueberry juice was separated using methanol and 0.02 mol·L^−1^ ammonium acetate solution (5:95) as a mobile phase [32]. The injection volume was 10 μL, and the flow rate was 1.0 mL·min^−1^. The column temperature was 25 °C, and the detection was performed at 230 nm. Sodium benzoate was determined by the same method described for potassium sorbate. The standard equations of citric acid, potassium sorbate, and sodium benzoate were y = 2201x + 11,657 (R^2^ = 0.9994, 2–100 mg·L^−1^), y = 43,051,062x − 202,507 (R^2^ = 0.9995, 0–400 mg·L^−1^), and y = 25,568,511x − 118,381 (R^2^ = 0.9996, 0–400 mg·L^−1^), respectively.

### 2.12. Storage Experiment of Concentrated Blueberry Juice

In order to explore the effects of food additives on blueberry juice, the storage of concentrated blueberry juices with these three food additives was carried out. Concentrated and original blueberry juices were stored in a 50 mL sterile polypropylene tube at 25 °C. The TAC, TPC, TSS, pH, and anthocyanins profiles of blueberry juice were determined at weeks 0, 1, 2, and 3.

### 2.13. Statistical Analysis

All experiments and analytical measurements were performed in triplicate. Experimental results were expressed as the means ± standard deviation (SD). All experimental data were graphed by Origin 2021.

## 3. Results and Discussions

### 3.1. Effects of DSs on Water Flux and Membrane Fouling

Figure 2 depicts the water fluxes with three DSs, which showed a general decreasing tendency, a sharp decline in the first 50 min, and a steady level towards the end of the running. This phenomenon was similar to a study on the FO concentration of beetroot juice by Trishitman et al. [33]. It was mostly brought on by internal and external concentration polarization on both sides of FS and DS, as well as membrane fouling. In FO processes, solutes in juice are adsorbed on the membrane surface and accumulated as a cake layer [34], making it more difficult for water to pass through the membrane. Permeate flow drastically decreased at the start of FO development, indicating that membrane fouling formed immediately [35]. This demonstrated that foulants, mainly suspended components, significantly blocked the FO membrane during the FO of grapefruit juice [36]. Similarly, large particle constitutes, such as polysaccharides and polyphenols in blueberry juice, occluded the membrane pores. The water flux of FO processes followed an order of citric acid >sodium benzoate >potassium sorbate when using these three food preservatives as DSs. Interestingly, when using citric acid as a DS, an abnormally higher water flux was shown compared to that of other DSs, even though citric acid (22.29 bar) had a lower osmotic pressure than that of sodium benzoate (53.24 bar) and potassium sorbate (22.63 bar) at 1 mol·L^−1^ [23,37]. Due to size exclusion and restricted diffusion, a larger solute size would reduce the effective porosity of the membrane, leading to an increase in internal concentration polarization (ICP) [38]. The size of H^+^ is much smaller than Na^+^ and K^+^; thus, a slighter ICP of citric acid might result in larger water flux. Therefore, citric acid showed good potential as a DS since it exhibits a large water flux. Moreover, as citric acid could be an antioxidant and acidulant, there is no need to remove it from the product.

Figure 3a shows the original membrane with some crystals on the surface. This might be the sodium bisulfite used as a preserve for the original membrane. In Figure 3b, it was found that fouled membrane surface was covered by blueberry juice residues. In Figure 3c, there were numerous crystals on the surface because of the reverse flux of potassium sorbate. However, there were fewer crystals on the membrane surface when sodium benzoate was used, as shown in Figure 3d. Blueberry juice contained lots of organic components, which could form a fouling layer on the surface of the FO membrane, as shown in Figure 3b–d. Due to the coarse porous structure of the TFC membrane’s supporting layer, some particles were easily adsorbed on the membrane or even into the membrane pores during the FO process [39], which also led to membrane fouling.

### 3.2. Effects of DSs on Characteristics of Blueberry Juice

#### 3.2.1. Effects on Physicochemical Properties

As shown in Table 1, the protective effect of citric acid on anthocyanins was the most significant, followed by sodium benzoate and potassium sorbate. This also indicated that after concentration by citric acid, the pH of blueberry juice decreased, while that with the others increased. This might be caused by the reflux of DS [40]. Moreover, anthocyanins were more stable in acidic conditions [41]. Therefore, the FO process with the DS of citric acid could protect polyphenols such as anthocyanins in juices.

The variation trends of TPC, total sugar, and TSS during concentrations with different DSs were all similar. TPC, total sugar, and TSS were increased by about 20% with citric acid, while these with two others were about 12% and 17%. Binding between phenolic compounds and other solutes such as polysaccharides might occur in the solution, resulting in larger particles adsorbed on the membrane [42], which causes a reduction in phenolic compounds. This also showed that FO had little influence on the chroma of blueberry juice. It has thus been proved that FO had little effect on phenols and anthocyanins, and the co-pigment effects of phenols could be protective to anthocyanins [43].

#### 3.2.2. Effects on Particle Size

Figure 4 shows the contribution of particles in juices in which the sizes between 600 and 2000 nm and 3000 and 8000 nm were relatively high. After concentration, the fraction of large particles in blueberry juice decreased. The percentage of 5 μm particles in blueberry juice was reduced from 5% to 1% because some solutes might be adsorbed on the membrane [44]. The reduction in particle size could improve the physical properties and the stability of juice [45]. Zhu et al. found that the particle size of tomato juice was reduced after hydrodynamic cavitation, which improved the stability of tomato juice [46]. Compared with the other two DSs, the size of large particles in juice concentrated with citric acid reduced the most significantly. Accordingly, it is beneficial to stabilize the blueberry juice using citric acid as a DS.

#### 3.2.3. Effects on the Anthocyanins Profile

Twelve anthocyanins were identified in blueberry juice, as reported in our previous study [47]. As shown in Figure 5, all peaks persisted steadily after concentrations, indicating that it made a slim difference to anthocyanins in blueberry juice during FO processes. Varied anthocyanins have different polarities due to different glycosides and substituents on them, which ultimately affects how well they adhere to FO membranes [48]. Delphinidin-3-O-galactoside, delphinidin-3-O-glucoside, and delphinidin-3-O-arabinoside increased slightly after concentration. These three anthocyanins were easily adsorbed on membranes because they contained delphinidin, which had numerous hydroxyl groups [49].

### 3.3. Effects of DS on Reverse Salt Fluxes and Juice Storage

In the FO process, the solutes of DS could reverse into the feed solution. In this study, citric acid, sodium benzoate, and potassium sorbate all reversed into the concentrated juice, and their reverse fluxes were 18.14 ± 2.81, 9.43 ± 0.05, and 5.54 ± 0.02 g·m^−2^·h^−1^, respectively. Correspondingly, the accumulations of food additives in the concentrated juice were 3.083 ± 0.477, 1.497 ± 0.008, and 0.869 ± 0.003 g/kg at the final measurement, respectively. FAO required that the contents of sodium benzoate and potassium sorbate in concentrated fruit and vegetable juice should not exceed 1.0 g/kg. Accordingly, in our study, the content of potassium sorbate in blueberry juice did not exceed the standard, but sodium benzoate exceeded. The citric acid in the juice was kept within the safety limit (3.0 g/kg, FAO). Similar findings demonstrated that the reverse salt flux has no significant effect on the FS, and even some concentrated products do not need to remove the solute of reverse penetration [14,23]. In general, it is safe to use food additives as a DS in the FO process of juices.

In Figure 6a, the trends of TAC all declined throughout storage, in which juice with citric acid decreased slowly. Anthocyanins could be easily oxidized by oxygen [50], but citric acid protected them because of its antioxidant effect. TAC in blueberry juice with sodium benzoate and potassium sorbate decreased rapidly, possibly because they raised the pH and made anthocyanin degraded more easily. Reque et al. [51] found anthocyanins suffered significant loss during 10 days of juice refrigeration, and the degradation rate reached 83% without antioxidants, indicating that citric acid could play an important role in the storage process of blueberry juice as an antioxidant. In Figure 6b, the variations of TPC in the juice were slight, and those containing citric acid and potassium sorbate were more stable. In other studies, TPC in juices without these food additives decreased more significantly [52]. This indicated that the pH of concentrated blueberry juices changed slightly during storage, as shown in Figure 6c. Figure 6d shows that the TSS of concentrated blueberry juice with all three food additives decreased slightly while that of original juice decreased significantly. In general, the stability of blueberry juice can be improved by using food additives as DSs because of the reverse flux to the FS.

## 4. Conclusions

In the study, the feasibility of using food additives as DSs for the FO process was demonstrated. Citric acid, sodium benzoate, and potassium sorbate were selected as DSs for blueberry juice concentration by FO. Citric acid exhibited higher water flux and more effective protection against anthocyanins than sodium benzoate and potassium sorbate. With citric acid as the DS, TAC in blueberry juice increased by about 26% and TPC by about 21%, which were higher than those with others. This indicated that citric acid had a particularly significant protective effect on anthocyanins. Citric acid could reverse into concentrated juice at the safety limit of FAO. These reversed food additives in concentrated juices could keep the juice more stable during storage. The protective effect of citric acid on the bioactive substances of juices demonstrated that food additives could be effectively applied in FO concentration processes of juices as DSs.

## Figures and Tables

**Figure 1 membranes-12-00808-f001:**
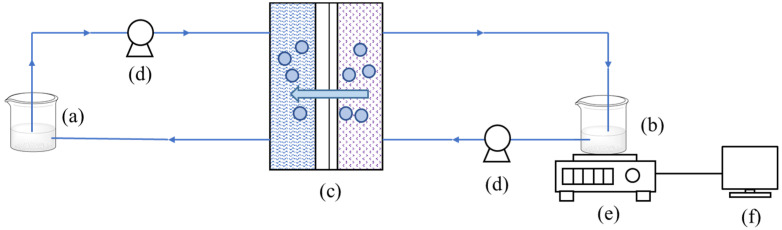
FO concentration process. (**a**) Draw solution; (**b**) Feed solution; (**c**) FO membrane module; (**d**) Peristaltic pump; (**e**) Digital balance; (**f**) Computer.

**Figure 2 membranes-12-00808-f002:**
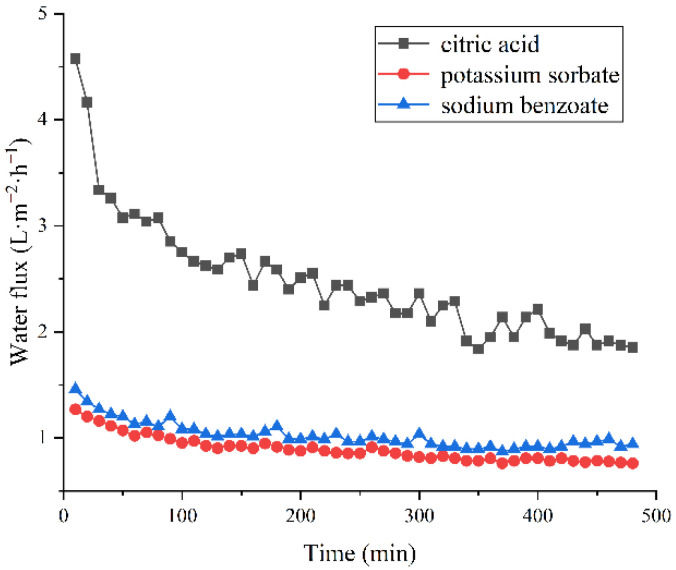
Water flux changes with different DSs. The concentration of DS: 2 mol·L^−1^; flow rate on both sides: 120 mL·min^−1^; membrane orientation: active layer towards the feed.

**Figure 3 membranes-12-00808-f003:**
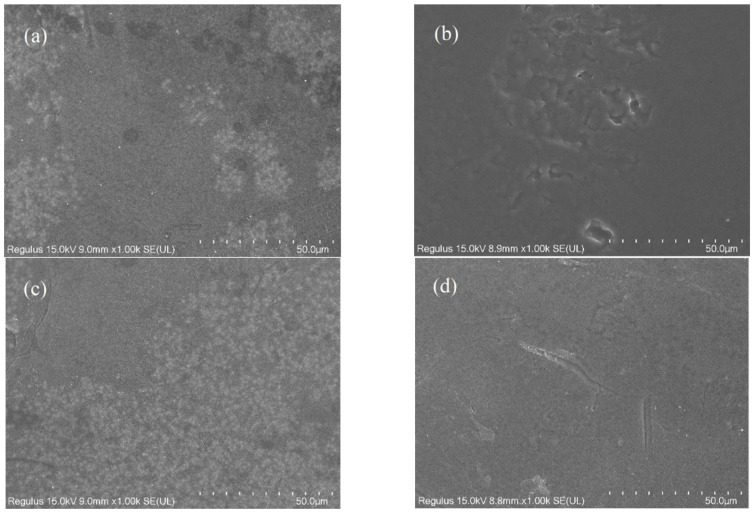
SEM images of the active layer of FO membrane. (**a**) Original membrane; (**b**) Citric acid; (**c**) Potassium sorbate; (**d**) Sodium benzoate.

**Figure 4 membranes-12-00808-f004:**
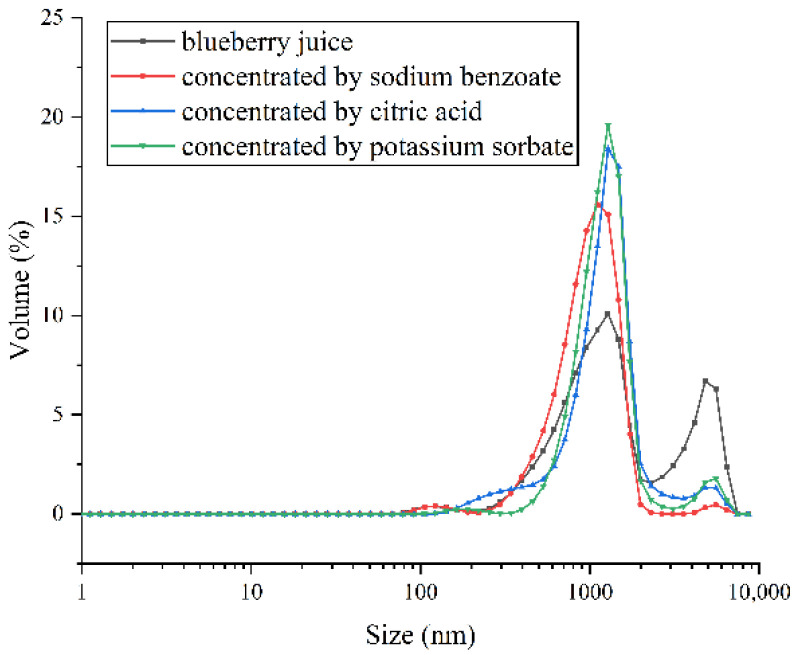
Size distribution of original and concentrated blueberry juices.

**Figure 5 membranes-12-00808-f005:**
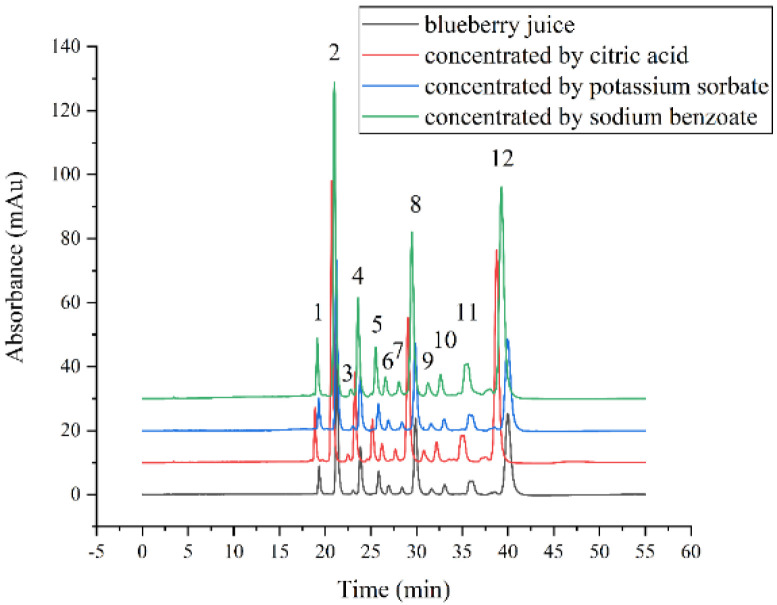
Liquid chromatogram profiles of original and concentrated blueberry juice. 1: delphinidin-3-O-galactoside; 2: delphinidin-3-O-glucoside; 3: delphinidin-3-O-arabinoside; 4: cyanidin-3-O-galactoside; 5: petunidin-3-O-galactoside; 6: petunidin-3-O-glucoside; 7: petunidin-3-O-arabinoside; 8: malvidin-3-O-galactoside; 9: peonidin-3-O-glucoside; 10: malvidin-3-O-glucoside; 11: malvidin-3-O-arabinoside; 12: delphinidin-3-O-xyloside.

**Figure 6 membranes-12-00808-f006:**
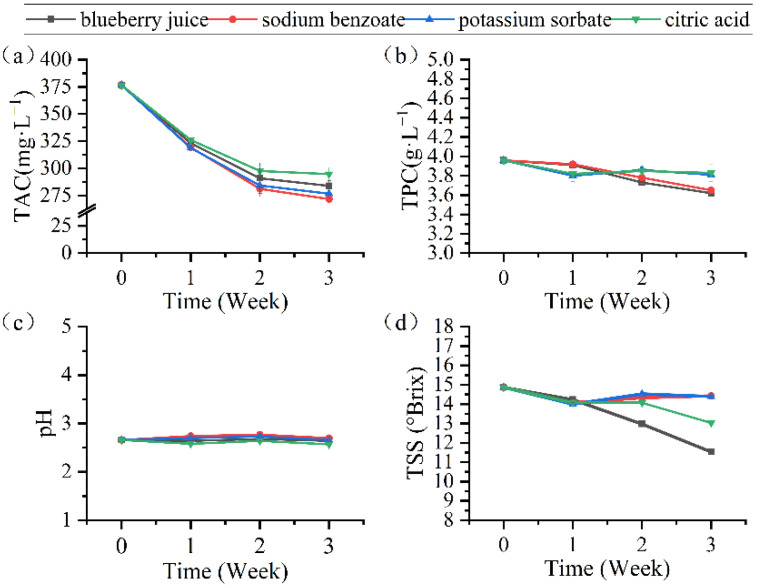
Changes of TAC (**a**), TPC (**b**), pH (**c**), and TSS (**d**) in concentrated blueberry juice with different DSs during storage.

**Table 1 membranes-12-00808-t001:** Physicochemical properties of juices concentrated with three draw solutions.

Draw Solution		Total Anthocyanin Content (TAC)/mg·L^−1^	pH	Total Phenolic Content (TPC)/g·L^−1^	Clarity/%	Total Sugar/g·L^−1^	Total Soluble Solid (TSS)/%	L *	a *	b *	ΔE
Citric acid	OS ^1^	599.49 ± 9.37	2.64 ± 0.01	5.39 ± 0.05	0.18 ± 0.01	125.04 ± 6.96	15.13 ± 0.05	32.51 ± 0.16	0.15 ± 0.22	−0.87 ± 0.17	0.47 ± 0.12
	CS ^2^	752.56 ± 29.04	2.36 ± 0.02	6.52 ± 0.10	0.17 ± 0.01	150.34 ± 2.64	18.14 ± 0.06	32.78 ± 0.36	−0.04 ± 0.20	−0.97 ± 0.15	
Potassium sorbate	OS	613.41 ± 12.20	2.74 ± 0.01	4.91 ± 0.03	0.12 ± 0.01	123.73 ± 3.16	15.08 ± 0.63	28.17 ± 0.78	0.34 ± 0.22	−0.07 ± 0.71	1.56 ± 0.87
	CS	735.31 ± 24.92	3.00 ± 0.01	5.43 ± 0.01	0.02 ± 0.05	138.79 ± 1.77	17.00 ± 0.10	28.67 ± 0.23	1.61 ± 1.01	−0.47 ± 0.37	
Sodium benzoate	OS	607.84 ± 10.97	2.70 ± 0.01	5.60 ± 0.05	0.06 ± 0.00	123.44 ± 3.65	15.47 ± 0.04	27.85 ± 0.09	0.12 ± 0.08	−0.17 ± 0.29	0.49 ± 0.25
	CS	716.10 ± 30.80	3.04 ± 0.01	6.53 ± 0.15	0.01 ± 0.00	146.39 ± 3.29	17.96 ± 0.20	27.88 ± 0.01	0.23 ± 0.20	−0.61 ± 0.15	

*: The value of L, a and b. 1: Original solution of blueberry juice. 2: Concentrated solution of blueberry juice.

## Data Availability

The datasets analyzed during the current study are available from the corresponding author on reasonable request.

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
