# Peer review of "Athermal Concentration of Blueberry Juice by Forward Osmosis: Food Additives as Draw Solution"

_membranes, 2022, doi:10.3390/membranes12080808_

Round 1

Reviewer 1 Report

The paper looks great now

Reviewer 2 Report

Dear Editor

I would like to inform you that the authors answer all of my comments but still the english language is lacks, therefore miner revision is needed

Regards

This manuscript is a resubmission of an earlier submission. The following is a list of the peer review reports and author responses from that submission.

Round 1

Reviewer 1 Report

The authors provide a study to evaluate the feasibility of using food additives as draw solution for the FO process. Although the authors provided a comparison about three different draw solutions, the novelty of this study is not enough since there are many previous studies that reported similar experiences and got similar key findings [1,2]. 

[1] An, X., Hu, Y., Wang, N., Zhou, Z., & Liu, Z. (2019). Continuous juice concentration by integrating forward osmosis with membrane distillation using potassium sorbate preservative as a draw solute. Journal of membrane science573, 192-199.

[2]Zhang, K., An, X., Bai, Y., Shen, C., Jiang, Y., & Hu, Y. (2021). Exploration of food preservatives as draw solutes in the forward osmosis process for juice concentration. Journal of Membrane Science635, 119495.

Specific comment:

Fig. 2, what is the reason for the similar water fluxes when using potassium sorbate and sodium benzoate as the draw solutes?

Fig. 2, What is the reason for the increased water flux from 50 min to 100 min when using potassium sorbate as the draw solute?

Reviewer 2 Report

Membranes

Athermal Concentration of Blueberry Juice by Forward Osmosis: Food Additives as Draw Solution

This study focuses on evaluating the athermal forward osmosis (FO) concentration process of blueberry juice using food additives as draw solution (DS). The authors were made a perfect research of intensive results with justifying the main comparison between three food additives as DS.

Please revise the paper within the following points before providing a suitable decision regarding the paper based on MINOR corrections

1.      Abstract: Use the present tense in the Abstract. For example, use (this study focuses on evaluating) instead of (This study was to evaluate an athermal). Also use (concentration processes are studied) instead of (concentration processes were studied). Please check the whole Abstract

2.      Abstract, line 21: these are same values (735.31±24.92 and 735.31±24.92 mg·L-1,) please check

3.      Introduction, lines 43-45: please cite proper references for this sentence. This is a suggestion to be included for RO process (not compulsory): https://doi.org/10.1016/j.jfoodeng.2017.06.020

4.      Introduction: please cite the sentence in lines 60-62

5.      Conclusions: please write a short sentence to represent the main aim of this study

Reviewer 3 Report

Dear Editor

Thank you for the invitation to review the manuscript entitles "A thermal Concentration of Blueberry Juice by Forward Osmosis: Food Additives as Draw Solution". The manuscript matches the scope of the journal and describes the forward osmosis (FO) as a concentration process for blueberry juice. The authors were proposed different food additives as draw solution in FO process. The authors found that food additives could be an excellent choice to be DS in FO  concentration process of juices.

 The general idea of the paper seems to be good. However, there are several major technical challenges that should be effectively addressed. This work can be accepted in Membranes after answer the major comments presented in the following report.

Best Regards

Comments to the authors:

1.      First of all, there are some grammatical errors and typos that should be corrected before publication

2.      The abstract has been briefly written without any information on the membrane used in the current work. All information on FO membrane should be move from section 2.2 to section 2.1.

3.      It is obvious that UF membrane has a broad use in food application, therefore the authors should present the benefit of using FO instead of UF in Introduction section.  

4.      In the introduction section the authors did not present the drawbacks and gaps of literature with the current work, and particularly, how the proposed approach aims at filling these gaps.

5.      Line 218 "porous structure of the TFC membrane" are the authors sure that TFC is porous!!!!!

6.      Line 246, "The particle size of blueberry juice represents the clarity of juice, and also affects the stability of blueberry juice [37]. After hydrodynamic cavitation, the particle size of tomato juice was reduced, which improved the stability of tomato juice [38].", also line 255 "Similar to tomato juice, the stability of blueberry juice would be improved." not clear sentences!!

7.      The authors should present the evidence that the research objective has been achieved in each section of the results and discussion.